# Ternary Metal-Alginate-Chitosan Composites for Controlled Uptake of Methyl Orange

**Bernd G. K. Steiger and Lee D. Wilson ***

Department of Chemistry, University of Saskatchewan, 110 Science Place, Thorvaldson Building, Saskatoon, SK S7N 5C9, Canada

**\*** Correspondence: lee.wilson@usask.ca; Tel.: +1-306-966-2961

**Abstract:** Three ternary metal composites (TMCs) with iron nitrate, aluminum nitrate, and copper nitrate (Fe-TMC-N, Al-TMC-N, Cu-TMC-N) were synthesized and their physicochemical properties were investigated. Characterization of the TMCs was achieved by elemental analysis (XPS), infrared (IR) spectroscopy and thermogravimetric analysis (TGA). The surface charge of the TMCs was estimated from the point-of-zero-charge (PZC), which depended on the type of metal nitrate precursor. The adsorption properties of the TMCs showed the vital role of the metal center, where methylene blue (MB) is a cationic dye probe that confirmed the effects of surface charge for effective methyl orange (MO) anion dye uptake. MB uptake was negligible for Al-TMC-N and Cu-TMC-N, whereas moderate MB uptake occurs for Fe-TMC-N (26 mg/g) at equilibrium. The adsorption capacity of MO adopted the Langmuir isotherm model, as follows: Al-TMC-N (422 mg/g), Cu-TMC-N (467 mg/g) and Fe-TMC-N (42 mg/g). The kinetic adsorption profiles followed the pseudo-second order model. Generally, iron incorporation within the TMC structure is less suitable for MO anion removal, whereas Cu- or Al-based materials show greater (10-fold) MO uptake over Fe-based TMCs. The dye uptake results herein provide new insight on adsorbent design for controlled adsorption of oxyanion species from water.

**Keywords:** ternary metal composite; methyl orange; dye adsorption; chitosan; alginate

## 1. Introduction

Dyes are organic substances that are excessively used in industry and research to stain various substances and textiles. Dyes are classified based on their application or chemical structure, where azo dyes comprise a large share of the synthetic dyes available due to their low cost, facile coupling reaction and wide range of substrates [1]. Azo dyes exhibit wide-spread use in biomedical applications, where they can disrupt the DNA of microorganisms, which can induce toxicity in organisms [2–4]. Depending on the structure, azo dyes are also carcinogenic, which can cause serious health effects at elevated concentration in their pristine form or after metabolism [5].

Methyl orange (MO) is an azo dye that is widely used across multiple industrial sectors, chemical laboratories and academic institutes, where its use in undergraduate organic labs relates to its facile synthesis [6]. The common use of MO, especially in the textile industry, results in MO laden wastewater, where remediation is needed to avoid environmental contamination. Additionally, the composition of such wastewater effluent can vary and pose challenges due to effects such as high salinity [7].

Efficient dye removal from industrial wastewater remains as an active field of research, where various dye removal techniques employ strategies such as electrochemical, oxidative, or photocatalytic decomposition processes. In contrast to chemical decomposition, adsorptive removal can be used to isolate dye molecules onto a solid phase that can be readily phase separated from solution media. The importance of

dye removal from such effluent is mirrored by continued research interest in the development of efficient removal methods. Katheresan et al. provided a comparative overview of various removal methods for specific dyes including average success rates (from microbial cultures (81.6%), advanced oxidation processes (97.3%) and electrochemical destruction (88.8%)) [8]. Selected removal methods can even achieve up to 100% removal, however, such methods require specific equipment with additional energy and material input requirements. Adsorptive removal offers a facile, low-cost and highly effective method for separation of dyes from wastewater. The additional benefit of adsorption is the association (physisorption or chemisorption) of the dye molecules onto a solid adsorbent via phase separation, where potential reuse of the dye and adsorbent regeneration can be achieved [9,10]. Research on the use of bio-wastes and bio-sorbent materials for wastewater treatment has become increasingly important [11]. For MO adsorption, a wide variety of materials from organic to inorganic materials can be used, with ca. 40% of adsorbents being composites and 14% polymers and resins [9]. For sustainable and facile adsorbent design, minimal chemical modification or resource inputs is desired, along with the utilization of abundant natural biomaterials such as polysaccharides.

Chitosan is derived from a natural biopolymer (chitin), which is a linear polysaccharide with β-(1→4) linkages. Chitin and chitosan can be distinguished by the degree of deacetylation (DAC), where the polysaccharide is typically considered chitosan with a DAC > 50% [12,13]. Due to the abundant amine groups of chitosan, it is considered cationic in nature ($pK_a \approx 6.5$ [14]) if the pH of the media is acidic (pH < $pK_a$). Chitin and chitinaceous biopolymers have seen widespread use for dye removal, especially anionic dyes such as MO [15–17].

Alginate on the other hand is the anionic form of the naturally occurring and edible polysaccharide (alginic acid), which contains mannuronate and guluronate residues [18]. Alginate has abundant carboxylate groups that makes it an excellent candidate to form polyelectrolyte complexes (PECs) with cationic (bio-)polymers such as chitosan for adsorption-based water remediation applications [19–21]. Additionally, such binary PECs can be further modified by additives to yield the formation of ternary metal or clay composites to achieve variable adsorption properties [22,23].

Kumar et al. reported the preparation of ternary metal composites (TMCs) that contain chitosan-alginate-aluminum for adsorptive removal of fluoride, chromate and dyes such as reactive black 5 (RB 5) [24]. More recently, Udoetok et al. [25] reported the utility of such TMCs for phosphate removal, in conjunction with a computational study to elucidate the mechanism of adsorption [26]. TMCs that contain aluminum and copper were reported to exhibit notable anion uptake capacity; especially fluoride, arsenate, sulfate and organic dyes such as RB 5. However, the metal oxidation state (Al(III) vs. Fe(III) vs. Cu(II)) for the cation species of the TMC and its role on the surface charge and the structure-function properties are not fully understood at present. Herein, this study attempts to highlight the role of the metal center and its composition on the adsorption properties of such ternary metal composites (TMCs) by studying several types TMC systems for the controlled uptake of MO.

It is posited, that the incorporation of the metal center (Al, Cu, Fe) and the potential formation of (interfacial) hydroxy groups ought to play a crucial role in the adsorption process [21]. Hence, this work will retain the composition of the biopolymer scaffold components (1:1 weight ratio) and employ the three different metal cations to achieve a side-by-side comparison of their utility for MO remediation. As well, methylene blue (MB) was used as a cationic dye probe to assess the structural role of different metal cations. As well, MO is contrasted with MB due to the variable charge state of these dyes at ambient conditions. Based on electrostatic considerations, the interaction of cation versus anion dyes is anticipated to provide insight on the dye exchange mechanism during the adsorption process to advance the state of knowledge in the field [27].

## 2. Materials and Methods

### 2.1. Materials

Sodium alginate with 120–190 kDa, chitosan (LMW, DDA ca. 82%), kaolinite, KBr (FT-IR grade, 99%+), methylene blue (98%), aluminum nitrate nonahydrate (98%+), iron(III) nitrate nonahydrate (98%+), $Cu(NO_3)_2 \times 2.5\ H_2O$ (98%), boric acid (99.5+%) were obtained from Sigma-Aldrich (St. Louis, MO, USA). Glacial acetic acid (99.7%), sodium hydroxide (97%), hydrochloric acid (36.5%) and sodium bicarbonate (ACS grade) were purchased from Fisher Scientific Canada. Water (Millipore, Burlington, MA, USA) for synthesis and analysis was purified to a resistivity of 18.2 $M\Omega \times cm$.

### 2.2. Methods

Dye Adsorption

Dye concentration was measured via UV-Vis spectroscopy (Thermo Fisher Scientific Spectronic 200E) (Waltham, MA, USA) at 464 nm (MO), 665 nm (MB). The spectrophotometric experiments were carried out in duplicate.

### 2.3. Point-of-Zero-Charge (PZC)

The $pH_{PZC}$ measurement was based on the pH shift method [28]. The sample (ca. 60 mg) was added to a fixed volume of 25 mL of 0.01 M NaCl solution to avoid any Ca-Alginate interaction at variable pH. After equilibrating for 48 h at 22 °C, the pH of each system in aqueous media was measured and the $pH_{PZC}$ was determined as the intersection of the final pH and a zero change in pH ($\Delta$ pH = 0) [28].

### 2.4. Thermogravimetric Analysis (TGA)

The weight loss profiles were obtained using a Q50 TA Instruments thermogravimetric analyser (TA Instruments, New Castle, DE, USA). Samples were heated in open aluminium pans at 30 °C for 5 min to allow for equilibration prior to heating at 5 °C/min to 500 °C.

### 2.5. FT-IR Spectroscopy

The FT-IR spectra were taken via a Bio-Rad FTS-40 (Bio-Rad Laboratories, Inc., Santa Clara, CA, USA) with the Kubelka-Munk method. The dried samples were mixed with FT-IR grade KBr in a 1:10 weight ratio (Sample: KBr) and thoroughly mixed. The diffuse reflectance infrared Fourier transformation (DRIFT) spectra were obtained at 295 K over a spectral range of 400–4000 $cm^{-1}$ with a resolution of 4 $cm^{-1}$. A minimum of 128 scans were recorded with a background spectral correction with KBr was performed.

### 2.6. X-Ray Photoelectron Spectroscopy

X-ray Photoelectron Spectroscopy (XPS) data was obtained using a Kratos (Manchester, UK) AXIS Supra system equipped with a 500 mm Rowland circle monochromated Al K-$\alpha$ (1486.6 eV) source, along with a hemi-spherical analyzer (HSA) and spherical mirror analyzer (SMA). A spot size of hybrid slot (300 μm × 700 μm) was used for data collection. All survey spectra were collected in the -5 eV–1200 eV binding energy range using 1 eV steps with a pass energy of 160 eV. An accelerating voltage of 15 keV with an emission current of 15 mA was used for the analysis.

### 2.7. pH Measurements

The pH was determined with a Mettler Toledo Seven Compact with Accumet 13-620-108A electrode.

### 2.8. Preparation of Composite Materials

The TMCs were prepared by dissolving 1 g chitosan in 100 mL (2% acetic acid solution) and 1 g sodium alginate in 100 mL in Millipore water, according to the synthetic procedure outlined by Kumar et al. [24]. Upon complete dissolution of the materials, the chitosan solution was slowly added with stirring to the alginate solution and stirred for 2 h at 22 °C. For the preparation of Al-TMC-N, 100 mL 1 M Al(NO₃)₃ solution was slowly added under stirring at 22 °C. Then, a 2 M NaOH (*aq*) solution was added until pH 6.5 was reached. The suspension was further stirred for 2 h and then left overnight (12 h) to precipitate. The precipitate was filtered and washed with copious amounts of water, followed by drying at 65 °C for 48 h.

The Cu-TMC-N and Fe-TMC-N materials were prepared in an analogous fashion (with 100 mL 1.0 M Cu(NO₃)₂ and 100 mL 1.0 M Fe(NO₃)₃ solution, respectively). The materials were ground into finely divided powders using a mortar and pestle.

### 2.9. Adsorption Studies

Equilibriums uptake studies were undertaken by adding ca. 20 mg TMC samples into a 10 mL MB solution of known concentration at pH 7. For samples with MO, a 10 mg sample was used in 10 mL solution of known MO concentration at pH 7. The equilibrium adsorption capacity was estimated by Equation (1).

$$Q_e = \frac{(C_0 - C_e)}{m} \times V \tag{1}$$

Herein, a volume of solution is shown as V, the adsorption capacity at equilibrium as $Q_e$, initial adsorbate concentration is $C_0$ at $t = 0$, and adsorbate concentration at equilibrium $C_e$ at variable time (t), and the mass of adsorbent (m).

The Sips isotherm model was used to investigate the adsorption parameters at equilibrium, according to Equation (2) [29].

$$q_e = \frac{Q_m(K_a C_{eq})^{1/n_s}}{1 + (K_a C_{eq})^{1/n_s}} \tag{2}$$

$K_a$ represents the equilibrium constant, $q_e$ the adsorption capacity at equilibrium, $Q_m$ the maximum adsorption capacity and $n_s$ refers to the heterogeneity coefficient.

In addition, the Freundlich isotherm model was used to investigate the equilibrium adsorption parameters, according to Equation (3) [30].

$$q_e = K_f C_e^{\frac{1}{n}} \tag{3}$$

$K_f$ is the Freundlich constant and n represents the Freundlich exponent.

The Langmuir isotherm model is commonly used to describe monolayer adsorption, where no lateral adsorbate-adsorbate interactions are assumed and the adsorbent surface sites are homogeneous in nature, according to Equation (4) [31].

$$q_e = \frac{q_m K_L C_e}{1 + K_L C_e} \tag{4}$$

$K_L$ represents the Langmuir constant, $q_e$ represents the equilibrium monolayer adsorption capacity, $C_e$ is the adsorbate concentration at equilibrium, $q_e$ the amount of bound adsorbate at variable adsorbate concentration.

### 2.10. Kinetic Studies

Herein, kinetic adsorption studies were undertaken by using 100 mg of TMC material in 250 mL MO solution with an initial concentration of ca. 230 mg/L. To investigate the adsorption kinetics, the profiles were fitted to a pseudo-first-order (PFO) model (Equation (5)) or a pseudo-second-order (PSO) model (Equation (6)) [32,33].

$$q_t = q_e(1 - e^{k_1 t}) \tag{5}$$

$q_t$ (mg/g) represents the adsorption capacity at variable time, $q_e$ (mg/g) the adsorption capacity at pseudo equilibrium, and $k_1$ is the PFO rate constant [32].

$$q_t = \frac{k_2 q_e^2 t}{1 + k_2 q_e t} \tag{6}$$

Here, the variables in Equation (6) are similarly defined as in Equation (5), whereas $k_2$ is the rate constant for PSO kinetic model.

Additionally, the intraparticle diffusion (IPD) model was employed, as defined by Equation (7) [34]:

$$q_t = kt^{0.5} + C \tag{7}$$

where C represents the intercept, $t^{0.5}$ is the square root of the contact time, and k is the kinetic rate constant.

### 2.11. Software

Model fitting and data processing was carried out using non-linear least squares procedure in Origin 2021b (64-bit) SR1 under academic license, data processing and writing in MS Excel (Office 365, Apps Enterprise).

## 3. Results and Discussion

In a previous study by Kumar et al., [24] the metal center of the TMC was inferred to play a critical role for the adsorption of pollutant anion species. Based on a TMC material that contained an Al(III) cation within its framework structure, inorganic anions such as fluoride and chromate or organic dye anions such as MO and RB 5 were removed either by chemisorption (fluoride) or physisorption (dyes) [27]. Furthermore, it was reported that the adsorption mechanism is generally governed by anion exchange, as evidenced by the change in adsorption capacity in the presence of multiple pollutants due to competitor ion effects [27]. Hassan et al. reported that variation of the biopolymer composition resulted in variable selectivity of sulfate adsorption for ternary metal composites that contain Cu(II) [25].

Herein, this study evaluates the role of switching the primary (metal ion) adsorption site for different metal cation species, while maintaining the composition of the biopolymer framework (chitosan, alginate) components. The structure-function relationship can be influenced by synthetic parameters (cf. Figure S1 in the Supplementary Materials) and the conditions must be carefully monitored. The prepared materials were investigated using MO and MB as dye probes to study the adsorption properties of the TMCs.

### 3.1. Characterization

The first step in the structural analysis of the materials was to investigate whether all elements from the precursors are incorporated into the material. Therefore, XPS was used to qualitatively analyze the composition of the TMCs.

### 3.1.1. XPS

The elemental analysis was accomplished via XPS, where the characteristic binding energies for each element affords unequivocal element identification [35]. Due to sufficient penetration of the samples by the X-rays and the finely powdered nature of the sample, this surface sensitive technique was used to approximate the bulk composition of the TMC materials (cf. Figure 1).

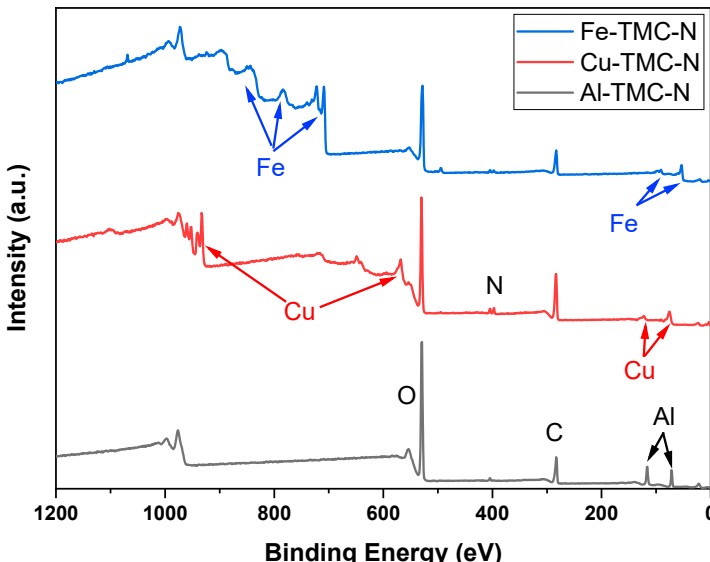

**Figure 1.** Elemental identification of Fe-TMC-N, Al-TMC-N and Cu-TMC-N.

Several elements (Fe, Cu and Al) were found in the respective XPS spectra, with an additional small Na signature in the Fe-TMC-N materials. Cu-TMC-N appears to also contain more nitrate than the other TMC materials.

A narrow scan of the Fe $2p_{3/2}$ orbital (cf. Figure S5, Supplementary Materials) showed the presence of FeOOH species within the material. The occurrence of Fe-O and zero-valent iron (Fe(0)) can be explained by X-ray induced reduction, whereas the synthetic pathway does not indicate the reduction of Fe(III) to Fe(II) or Fe(0). The narrow scan of the Cu $2p_{3/2}$ orbital (cf. Figure S5, Supplementary Materials) indicates the presence of $Cu(OH)_2$ species within the materials. Similarly to Fe, X-ray induced reduction to Cu(I) in the form of $Cu_2O$ was detected [35]. In contrast to the aforementioned metals, the oxidation state of Al is notoriously difficult to determine based on spectral overlap of XPS bands. Hence, the specific oxidation state of Al was not assigned herein. IR spectroscopy was used as a complementary method to investigate the presence or absence of related functional groups for the TMC materials.

3.1.2. IR Spectroscopy

Herein, the spectra of all three materials were obtained and normalized, along with alginate and chitosan, as noted in Figure 2. The absence or presence of expected bands including a potential spectral shift provide an indication as to how the functional groups may interact together within the composite structure.

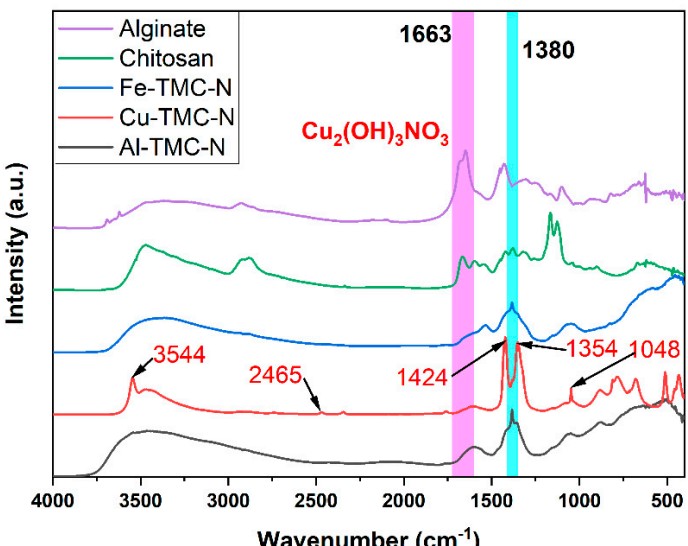

**Figure 2.** FT-IR spectra of Al-TMC-N, Cu-TMC-N and Fe-TMC-N with highlighted $Cu_2(OH)_3NO_3$ bands in the Cu-TMC-N spectrum.

The characteristic IR bands for alginate and chitosan were observed between 3500–2800 $cm^{-1}$ (O–H and C–H stretching, respectively). The bands are evident for chitosan, but more strikingly in the case of alginate, along with C=O stretching between 1700–1663 $cm^{-1}$. This trend was assigned, which was reduced in intensity for all three materials, indicative of coordination within the composite structure by this abated intensity. The band at ca. 1595 $cm^{-1}$ (found in chitosan, Al-TMC-N and Cu-TMC-N) was assigned to the N–H bending of amine groups that originate from chitosan. The absence of this band in Fe-TMC-N could mean greater chelation between the free amine groups with the iron sites of this TMC. The band around 1167 $cm^{-1}$ was assigned to C–N stretching of the amine group. The C–O–C bond, indicated by C–O stretching at ca. 1101 $cm^{-1}$ is most visible for alginate, but also likely overlaps with other bands in the biocomposites [27]. The expected characteristic nitrate band at 1383 $cm^{-1}$ was present in Al-TMC-N and Fe-TMC-N, but not evident for Cu-TMC-N. However, in Cu-TMC-N, the bands at 3544 $cm^{-1}$, 2465 $cm^{-1}$, 1424 $cm^{-1}$, 1354 $cm^{-1}$ and 1048 $cm^{-1}$ concur with the position of accompanying vibrational bands for $Cu_2(OH)_3NO_3$ [36]. This indicates that the copper hydroxy nitrate complex undergoes phase separation within the TMC material, unlike in the other materials, where nitrate most likely functions solely as a counterion for charge neutralization, which can also interact with the biopolymer framework. Thermal analysis by monitoring weight loss profiles has been reported to provide useful insight on the characterization of biopolymer composites [25–27], as described in the next section.

### 3.1.3. TGA and DTGA

To investigate the thermal stability of the materials, their decomposition profiles were analyzed by TGA and the decomposition profiles were plotted via their decomposition profiles and derivative profiles (cf. Figure 3A,B).

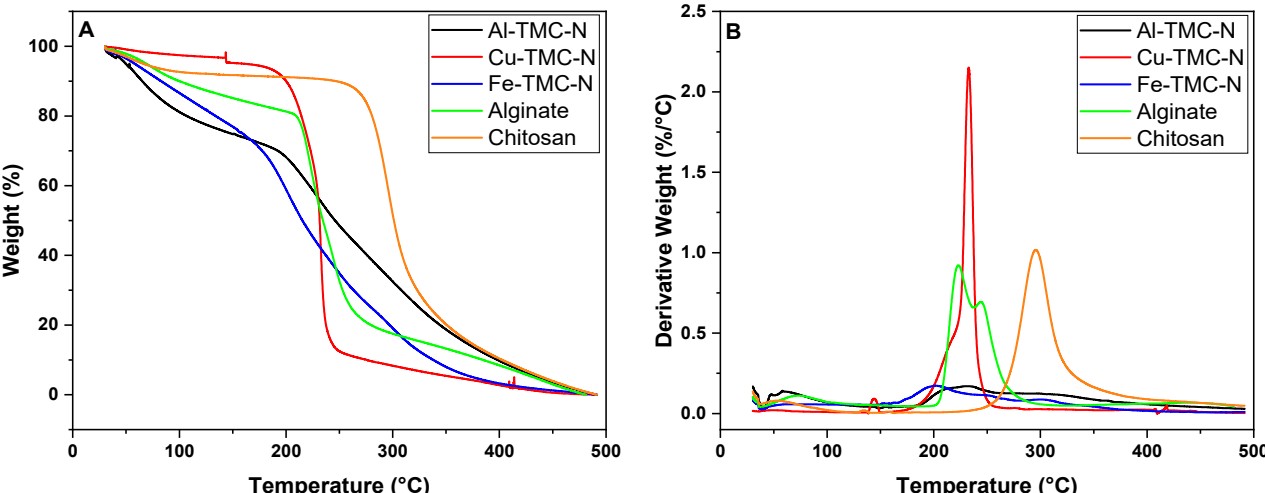

**Figure 3.** Thermogravimetric analysis (**A**) and derivative thermogravimetric analysis (**B**) of precursor biopolymers and TMCs: Al-TMC-N, Cu-TMC-N, Fe-TMC-N, alginate and chitosan.

The thermogravimetric analysis of sodium alginate shows initial water loss (up to 20%) up to ca. 200 °C, where sodium alginate exhibits its decomposition onset (see also first and second decomposition event in Figure 3B), which can be assigned decarboxylation and general decomposition [37]. In the temperature range of 200–260 °C, alginate loses approximately 60% of its weight. By contrast, chitosan shows water loss around 30–50 °C (ca. 10%), and a broad decomposition event from ca. 250–350 °C (ca. 80% weight loss), with a maximum derivative weight loss occurs near 300 °C. In comparison, both Al-TMC-N and Fe-TMC show increased water loss until ca. 100 °C. Cu-TMC-N [38], however, Fe-TMC does not exhibit such water loss and exhibits weight loss starting from ca. 180–250 °C with ca. 90% weight loss. This can also be corroborated by the derivative weight loss profile (shoulder at ca. 200 °C and main event at ca. 225 °C, Figure 3B).

Fe-TMC-N shows the earliest onset for the decomposition event near 200 °C, followed by the Al-TMC-N system around 230 °C, along with a decomposition event near 300 °C. For the Al-TMC-N and Fe-TMC-N materials, they lose approx. 60–70% of their weight starting from ca. 170 °C to 500 °C. In contrast to pristine chitosan or sodium alginate, the change in degradation profile may relate to incorporated acetic acid/acetate through the synthetic procedure, due to Lewis-acid and nitrate incorporation [39].

The different decomposition events and variable level of nitrate incorporation within the structure for the Cu-TMC-N compared to both Al-TMC-N and Fe-TMC-N is supported by the IR spectra of the composites (cf. Figure 2). The spectra provide support for changes in the surface chemistry and charge of the materials, which can be further investigated by estimating the point-of-zero-charge for the TMC materials.

### 3.1.4. PZC

The pH-shift method is a facile and valuable tool to investigate the approximate surface charge based on the condition where the materials have no resulting effective charge. The pH-shift method may be a suitable option when the measurement of the $\zeta$-potential is not feasible [28,40,41]. The initial pH was set as the abscissa and $\Delta$ pH after 48 h as the ordinate, which were plotted to find the point-of-zero-charge (PZC) of the materials (cf. Figure 4).

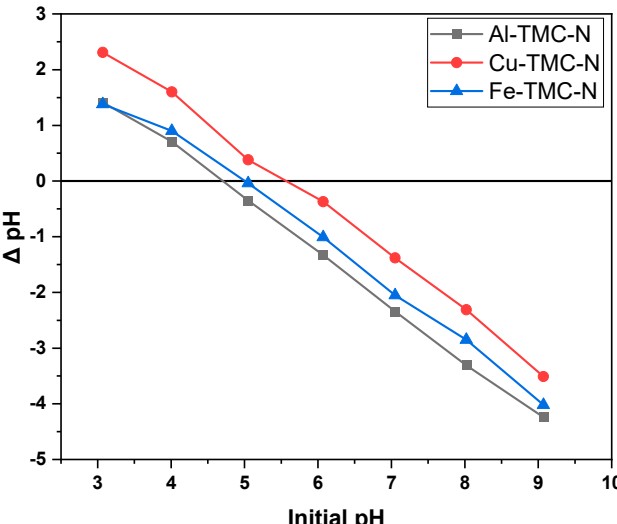

**Figure 4.** Determination of the PZC of the TMC materials via Δ pH of the starting solution and pH after 48 h with adsorbent.

The PZC of the materials was expected to be near pH 5 for the Al-TMC-N material, as reported elsewhere [27]. Here, the PZC of Al-TMC-N was found to be at pH 4.7 and is in approximate agreement, which can be explained by pH determination error of 0.2 pH units. Interestingly, the PZC of the other nitrate containing materials, the PZC were higher (pH 5.0 and 5.7 for Fe-TMC-N and Cu-TMC-N, respectively). This trend concurs with a stronger coordination of the counterion present [27], and is also shown for $Cu_2(OH)_3NO_3$ formation, based on IR spectral evidence (cf. Figure 2). The lower affinity for the coordination of nitrate within the Fe-TMC-N framework and different biopolymer interactions may account for the intermediate PZC value of the Fe-TMC-N composite.

The effect of the different metal coordination sites within the composite materials can also be supported by their respective adsorption behavior with a particular adsorbate system. The positive surface charge is expected to electrostatically repel adsorbates with a positive charge. Thus, MB was used as a cation dye probe to assess the adsorption properties at equilibrium conditions. By contrast, MO was used as an anion dye probe to evaluate the adsorption properties of the TMC materials under kinetic and equilibrium conditions.

### 3.2. Adsorption Studies

Adsorption of either cationic or anionic dyes can enable the distinction between primary versus secondary adsorption sites, along with the role of the biopolymer backbone. Herein, two types of dyes (MB and MO) were used to survey the nature of the active adsorption sites in the TMC composites. Additionally, a phenolic dye probe was employed to gain insight on the role the biopolymer adsorption sites by employing *p*-nitrophenol (PNP) at several different pH conditions (8.4, 6.9 and <6.3). These conditions correspond to variable charge state of PNP since the p$K_a$ for PNP is ca. 7.16 according to Aktaş et al. [42].

### 3.2.1. Methylene Blue (MB) Dye Adsorption

Methylene blue is a cationic dye that was expected to show negligible adsorption capacity towards the TMC materials, based on repulsive electrostatic interactions with the active metal ion center, whereas little to no contribution arises from the organic biopolymer fraction of the composites. The adsorption parameters of MB were evaluated by use of the Sips isotherm model only, which, depending on the n-value exponent provides an account of Freundlich or Langmuir adsorption behavior. In turn, the Sips

adsorption model may be more appropriate to account for general adsorption properties, according to the results in Figure 5.

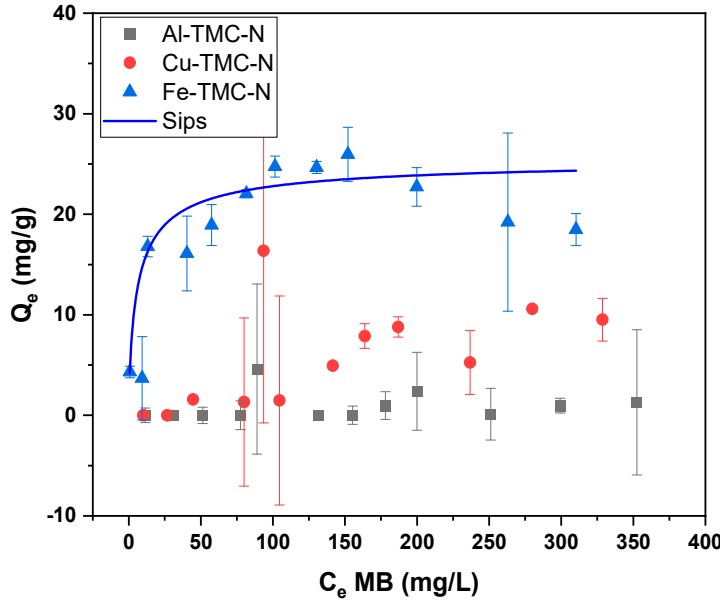

**Figure 5.** Equilibrium uptake profiles of MB onto three types of TMC materials, where only Fe-TMC-N was fitted by the Sips isotherm model (blue line) due to limited adsorption displayed by the other TMCs that contain Al(III) and Cu(II).

For both Al-TMC-N and Cu-TMC-N, no fit was attempted as the materials do not appear to have any appreciable adsorption of MB. This would indicate that these materials are potentially more suitable candidates for MO dye removal since it has a negative charge, as compared with the cationic MB dye.

The Fe-TMC-N shows some adsorption capacity with MB, where a maximum capacity of $25.6 \pm 2.7$ mg/g, $n = 0.75 \pm 0.2$, $K = 0.16 \pm 0.09$, with an $R^2 = 0.96$. This is in contrast to the other TMC materials and indicates that iron coordination within the biopolymer matrix results in variable surface chemistry, despite its PZC suggesting otherwise (cf. Figure 4). The variable charge repulsion may originate from the coordination of iron and iron oxide sites within the TMC materials, where some cationic species can interact with the iron (hydr)oxide surface sites.

### 3.2.2. Methyl Orange (MO) Dye Adsorption

The difference in the type of incorporated Lewis acid may affect the uptake characteristics of the target dye (methyl orange). Herein, the dye adsorption kinetics was investigated at low dosage to further highlight potential differences between the TMC adsorbent materials, as illustrated in Figure 6.

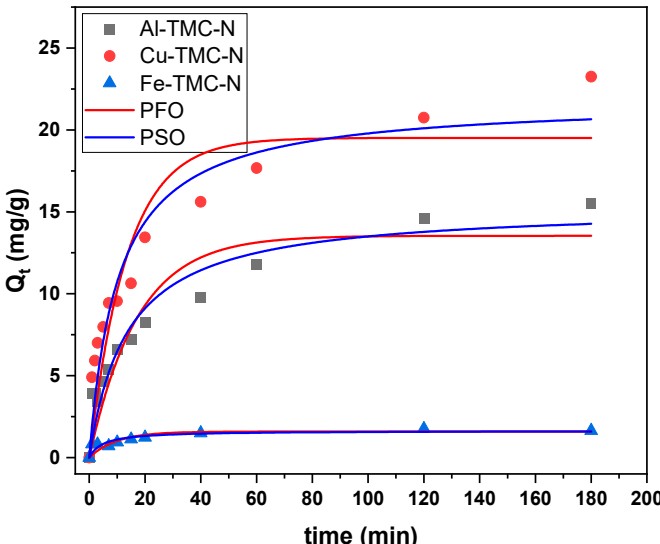

**Figure 6.** Adsorption kinetic profiles of MO adsorption onto the TMC materials at pH 7 (without buffer): Fe-TMC-N, Cu-TMC-N and Al-TMC-N at 295 K.

Interestingly, the adsorption kinetics for Fe-TMC-N are very low at these conditions (low dosage, low concentration) and show $q_t$ = 1.6 ± 0.2 mg/g for both PSO and PFO. However, according to the $R^2$ value, the adsorption profile is better described by the PSO kinetic model (0.71 for PFO vs. 0.8 for PSO). By contrast, the Cu-TMC-N composite reveals the highest uptake $q_t$ = 21.8 ± 1.4 (mg/g), whereas Al-TMC-N was 15.4 ± 1.1 (mg/g) at the same conditions. In both cases, the PSO kinetic model is obeyed for these systems with variable k values for the composites: Al-TMC-N (k = 0.005 ± 0.001), Cu-TMC-N (k = 0.004 ± 0.001) and Fe-TMC-N (k = 0.12 ± 0.07). This trend correlates to the previously shown MB adsorption data, where the Fe(III) containing material showed some uptake, indicating that it might be less suitable for adsorption of anionic dye species [27].

To confirm the observed differences in kinetic profiles, the intraparticle diffusion (IPD) model was used (cf. Figures S6 and S7 and Tables S1–S3 in Supplementary Materials). To further understand the kinetic uptake profiles, one single linear fit of all three components (cf. Figure S6) was compared versus a two-line fitting over two regions (cf. Figure S7, Supplementary Materials). Whereas Cu-TMC-N reveals a best-fit over the two regions, Fe-TMC-N shows only one region with negligible uptake overall. Al-TMC-N can be described with both single linear fit and two fits. For the case of two regions, the results may indicate contributions arising from film diffusion and subsequent intraparticle diffusion (IPD) [34]. Some key observations are outlined, as follows: (i) For Al- and Cu-TMC-N materials, the slopes of region 1 are generally greater than for region 2; and (ii) The observed trends noted for the results in Figure S6 parallel those observed in Figure 6. The variable slopes noted in regions 1 and 2 for the various TMC materials can be explained by the respective metal systems, variable coordination geometry, and HLB profiles, which concur with the equilibrium adsorption results for the TMC/MO systems reported in the following section [43].

It can be postulated, that Fe-TMC-N may yield a relatively low equilibrium adsorption capacity, according to its value of $q_t$. By comparison, Cu- and Al-TMC-N composites are likely to exhibit greater dye adsorption capacity. To gain further insight on the equilibrium uptake properties for the TMC composites, equilibriums adsorption capacity of MO was determined, according to the adsorption isotherms illustrated in Figure 7.

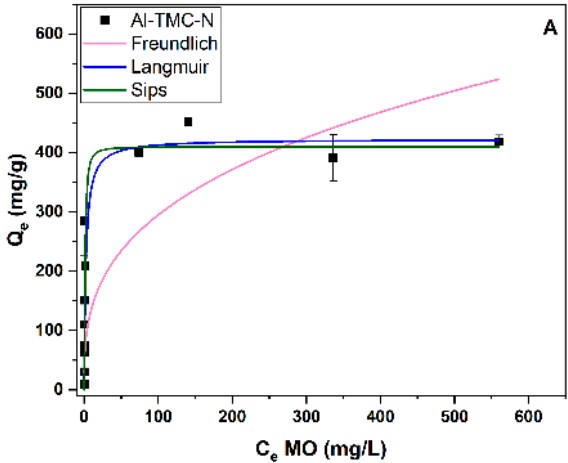

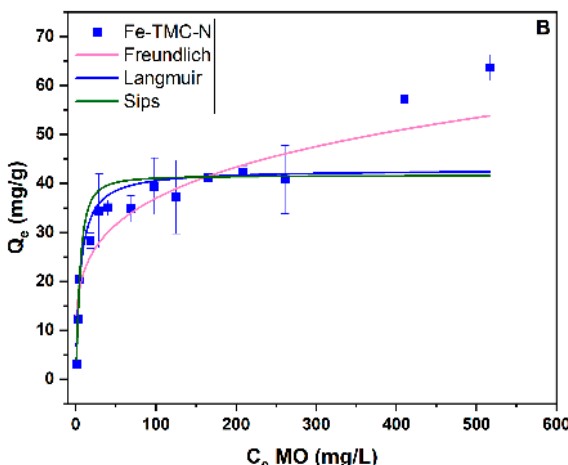

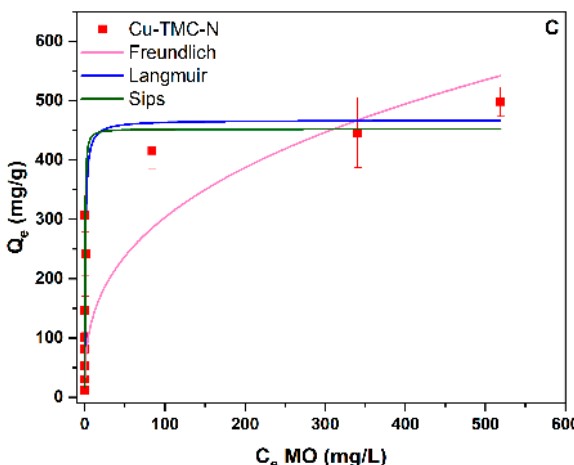

**Figure 7.** Methyl Orange adsorption profiles and best-fits by various models (Freundlich, Langmuir, and Sips) for the TMC materials: Al-TMC-N (**A**), Fe-TMC-N (**B**), Cu-TMC-N (**C**).

The sharp rise in the slope indicates chemisorption-like behavior of MO onto the metal centers for all materials, where the overall adsorption for the Fe-TMC-N material is considerably lower, as noted by a less pronounced rise. The adsorption profiles conform to the Langmuir isotherm model (see Tables 1, S2 and S3 in Supplementary Materials).

**Table 1.** Langmuir isotherm parameters for the TMC materials at 295 K.

|  | **Al-TMC-N** | **Cu-TMC-N** | **Fe-TMC-N** |
|---|---|---|---|
| q (mg/g) | $422 \pm 26$ | $467 \pm 81$ | $42 \pm 0.8$ |
| K | $0.4 \pm 0.1$ | $1.2 \pm 0.4$ | $0.18 \pm 0.02$ |
| R² | 0.97 | 0.88 | 0.97 |

The visual discoloration of the solution after achieving equilibrium during the adsorption process (cf. Figure S2 and S3, Supplementary Materials) aligns with the uptake capacities calculated by the Langmuir isotherm model. The Langmuir isotherm model provides favourable best-fit results versus the Sips and Freundlich models.

Iwuozor et al. provided a comprehensive comparison of MO uptake of various components, where the results obtained in this study exceed the adsorption capacities of the other types of sorbents such as modified coffee waste (ca. 60 mg/g), biochar (41–182 mg/g). By comparison, the adsorption capacity does not reach the MO uptake capacity of organosilica (1708 mg/g) and starch-modified ZnMgAl LDH (1555 mg/g) [9]. Therein, the mean adsorption capacities of many biosorbents and composites (250–292 mg/g) were exceeded and are comparable to those reported for clays, minerals and polymer resins (cf. Table 9 from Ref. [9]).

The presented systems and measured adsorption capacity values are representative for a range of dyes with variable charge state with MO as the dye probe. The systems studied did not introduce effects due to salinity, competitor ions, or pH effects. To attain a best-case scenario and an upper threshold for the adsorption capacity, the use of ideal conditions may not reflect the trends for environmental water samples or complex media of multicomponent systems. To briefly and succinctly test how the adsorption capacity decreases, a borate buffer system at three different pH values and one carbonate buffer system at one pH value was employed (cf. Figure S4, Supplementary Materials). Two main trends were observed: $Q_e$ decreased with increasing buffer concentration and $Q_e$ increases with decreasing pH. Therefore, it can be assumed that the adsorption performance of the TMCs toward MO depends on various factors. For example, competitive effects from other anions in solution being a major contributor, aside from pH conditions, which can provide an account for the trends observed by invoking an anion-exchange mechanism for this particular adsorbate-adsorbent system [27,38].

### 4. Conclusions

Herein, three ternary metal composites (TMCs) were prepared (Fe-TMC-N, Al-TMC-N and Cu-TMC-N), where pH adjustment during composite formation is a key synthetic parameter, for Fe(III) and especially Cu(II) systems that may form (hydr)oxy species (cf. Figure S1, Supplementary Materials). Structural characterization via IR spectroscopy (cf. Figure 2) indicates the formation of $Cu_2(OH)_3NO_3$, which is corroborated by XPS (cf. Figure S5 in Supplementary Materials), whereas FeOOH is the dominant iron species for Fe-TMC-N. By contrast, Al- and Cu-species strongly interact with the biopolymer backbone (cf. Figures 2, 3 and S5), as supported by other independent studies [24,26,27]. The PZC results also reveal that the metal center influences surface charge and adsorption properties, according to the nature of the metal interactions with the functional groups (e.g., interfacial -OH) of the biopolymer [[27,44–47]]. Methylene blue (MB) was used as a cationic dye probe to evaluate the role of the metal centers of the TMCs, where no discernible adsorption of MB occurred for Cu- and Al-TMC-N composites, while measurable uptake was noted for Fe-TMC-N. These trends in uptake for MB reveal differences in surface chemistry and the role of coordination of the metal centers with the biopolymer framework. The anion dye adsorption properties were evaluated using methyl orange (MO), where the Langmuir model provided the best-fit results. The MO uptake capacity for the TMC materials are listed: Fe-TMC-N (42 mg/g), Al-TMC-N (422 mg/g), and Cu-TMC-N (467 mg/g). The adsorption kinetics for MO followed the PSO

kinetic model, where variable uptake capacity values are given: Fe-TMC-N (1.6 mg/g), Cu-TMC-N (21.8 mg/g), and Al-TMC-N (15.4 mg/g).

Herein, we demonstrate that the metal cation directly interacts with the carboxyl, hydroxyl and amine groups of the TMC material, which may also undergo ligation with the metal centers. Furthermore, the interfacial hydroxy groups surrounding the metal cation play thereby a crucial role in the adsorption process by altering the effective charge and hydration of the active metal center. This trend is revealed by the lower MO adsorption capacity for Fe-TMC-N versus Cu- and Al-based TMCs, where it can be concluded that Cu-/Al-TMCs possess superior anion adsorption over Fe-based systems.

**Supplementary Materials:** The following are available online at www.mdpi.com/article/10.3390/surfaces5040031/s1, Figure S1: Visual difference between the prepared materials with careful (left) neutralization step for successful materials preparation vs. metal oxide precipitation and lack of metal incorporation into the framework (right); Figure S2: After the kinetics study ($C_0$ ca. 230 mg/L), where 100 mg material were submerged in 250 mL solution. No filtration was used to get the optimum performance; withdrawal of material was negligible; Figure S3: Adsorption isotherm studies with methyl orange. Starting solutions in the foreground while Al-TMC-N and Cu-TMC-N (Fe-TMC-N not shown for clarity and lack of visual difference) are after adsorption. Solutions from 10–1000 mg/L were tested; Figure S4: Adsorption of MO in borate buffer solutions of varying strength (10–50 mM) and carbonate buffer (10–50 mM) with determined pH before adsorption and after adsorption.; Figure S5: XPS narrow scan of Fe $2p_{3/2}$ (A), Cu $2p_{3/2}$ (B) and Al $2p_{3/2}$ (C); Figure S6: Single linear fit and intraparticle diffusion model of all three TMC materials; Figure S7: Linear fit of two regions and intraparticle diffusion model of all three TMC materials; Table S1: Intraparticle diffusion model according to Weber and Morris for all three materials and MO adsorption kinetics; Table S2: Freundlich isotherm parameters for all three TMC materials; Table S3: Sips isotherm parameters for all three TMC materials.

**Author Contributions:** Conceptualization, L.D.W.; Methodology, B.G.K.S. and L.D.W.; writing, B.G.K.S.; review and editing, L.D.W. Supervision, L.D.W.; Funding acquisition, L.D.W. All authors have read and agreed to the published version of the manuscript.

**Funding:** L.D.W. acknowledges the support provided by the Government of Canada through the Natural Sciences and Engineering Research Council of Canada (Discovery Grant Number: RGPIN 04315-2021) and the Ministry of Agriculture and the Canadian Agriculture Partnership, through the Agriculture Development Fund (Project # 20170247).

**Institutional Review Board Statement:** Not applicable.

**Informed Consent Statement:** Not applicable.

**Data Availability Statement:** The data presented in this study are available in this article and the accompanying Supplementary Materials herein.

**Acknowledgments:** The authors wish to acknowledge Heng Yang for technical assistance with the adsorption experiments. The Saskatchewan Structural Science Centre (SSSC) is acknowledged for providing facilities to conduct this research. Funding from Canada Foundation for Innovation, Natural Sciences and Engineering Research Council of Canada and the University of Saskatchewan support research at the SSSC.

**Conflicts of Interest:** The authors declare no conflict of interest.

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
