# Peer review of "Ternary Metal-Alginate-Chitosan Composites for Controlled Uptake of Methyl Orange"

_surfaces, doi:10.3390/surfaces5040031_

Round 1

Reviewer 1 Report

The manuscript is well written and covers an interesting topic. I suggest the manuscript for publication after making the following changes;

1-     Line 6 of the abstract “Methylene blue (MB) was used as dye probe the surface chemistry of the TMC materials”, is confusing. Rewrite it.

2-     Also add the optimized conditions for adsorption in the abstract.

3-     Mention the hazardous effects of MO dye in the introduction section.

4-     Mention the novelty of your work.

5-     Add numbering to the sub-headings, i.e. 2.1, 2.2, ….

6-     In the results and discussion section, add the results obtained by the previous reports to make a comparison.

7-     There are too many grammatical and typing mistakes in the manuscript, reconsider it.

Author Response

Author Response to Reviewer Comments on MS ID:  surfaces-1921009

Ternary metal-alginate-chitosan composites for controlled uptake of methyl orange

The author comments are listed below in a point-by-point fashion, as listed in blue font below.

Reviewer #1 Comments

The manuscript is well written and covers an interesting topic. I suggest the manuscript for publication after making the following changes;

  • Line 6 of the abstract “Methylene blue (MB) was used as dye probe the surface chemistry of the TMC materials”, is confusing. Rewrite it.

Response: This sentence has been rephrased.

Also add the optimized conditions for adsorption in the abstract.

Response: We aimed to stay within the 200 word limit for the abstract and revised the abstract accordingly.

  • Mention the hazardous effects of MO dye in the introduction section.

Response: We have added additional references and briefly elaborate on the toxicity of MO in the introduction.

  • Mention the novelty of your work.

Response: We have revised the introduction accordingly.

  • Add numbering to the sub-headings, i.e. 2.1, 2.2, ….

Response: We added sub-headings.

  • In the results and discussion section, add the results obtained by the previous reports to make a comparison.

Response: For brevity, we made the reference in the discussion and provided few select examples for a comparison of MO adsorbents to put the proposed materials in perspective.

  • There are too many grammatical and typing mistakes in the manuscript, reconsider it.

Response: The manuscript has been revised to to address language, clarity and syntax to address the reviewer comment.

In summary, the Authors acknowledge Reviewer #1 for the constructive and insightful comments, along with the opportunity to improve the overall quality of the manuscript submission.

Reviewer 2 Report

In this study, three ternary metal composites were prepared and employed as adsorbents to remove dyes from aqueous solution. Characteristics of the composites and their adsorption efficiency towards MO and MB were investigated. This study well matches the requirements of Surface. I recommend to publish it after a minor revision. The detailed comments are presented as follow:

1. The full name of "MO" in line 18 should be given.

2. methyl orange in lines 58, 61, 157 should be MO.

3. Please confirm the unit in line 78.

4. TG curve should be given and analyzed.

5. Adsorption fitting using more isotherm models are recommended.

Author Response

Author Response to Reviewer Comments on MS ID:  surfaces-1921009

Ternary metal-alginate-chitosan composites for controlled uptake of methyl orange

The author comments are listed below in a point-by-point fashion, as listed in blue font below.

Reviewer #2 Comments

In this study, three ternary metal composites were prepared and employed as adsorbents to remove dyes from aqueous solution. Characteristics of the composites and their adsorption efficiency towards MO and MB were investigated. This study well matches the requirements of Surface. I recommend to publish it after a minor revision. The detailed comments are presented as follow:

  1. The full name of "MO" in line 18 should be given.

Response:  We addressed the comments, see revised manuscript

  1. methyl orange in lines 58, 61, 157 should be MO.

Response: We addressed the comments, see revised manuscript

  1. Please confirm the unit in line 78.

Response: We addressed the comments, see revised manuscript

  1. TG curve should be given and analyzed.

Response: We now include TG curves and revised the manuscript accordingly by analysing TG and DTG curves.

  1. Adsorption fitting using more isotherm models are recommended.

Response: We appreciate the feedback and now included Freundlich and Langmuir isotherm models. The Langmuir isotherm model turned out to yield the best fit and we revised the manuscript accordingly.

In summary, the Authors acknowledge Reviewer #2 for the constructive and insightful comments, along with the opportunity to improve the overall quality of the manuscript submission.

Reviewer 3 Report

Overall Comment

The paper entitled “Ternary metal-alginate-chitosan composites for controlled uptake of methyl oarnge”, showed some promising findings. Nonetheless, authors need to address some comments before ready for acceptance.

Abstract

Abstract is fine, however authors should include some numerical findings in for the physicochemical properties of the adsorbent in this case. Line 16 – 20 sounded oddly written. Why Fe-TCM-N showed such a low result, can you include some explanation there?

Introduction

Do add in the acute and long term health effects on human health in the writeup.

Line 24-39, definition of dyes and adsorption are too vague in this case. Please refer to the manuscripts below for improvement

1.      Engineered macroalgal and microalgal adsorbents: Synthesis routes and adsorptive performance on hazardous water contaminants

2.      Efficiency of various recent wastewater dye removal methods: A review

Some in-text citations are missing the year info.

Methodology

First of all, sub headings are missing for writeup

I have through the document for PZC measurement, why was the testing solution changed from calcium chloride to water? Based on what basis?

For the preparation of composite materials, is this the first research work done to synthesize in such a manner|? If no, please include the ref.

Generally, there are so many isotherms available, why particularly SIPS isotherm was chosen? Has the data been fitted with other isotherm as well, please include.

Results and Discussion

First of all, sub headings are missing for writeup

For the TGA, the thermograms are missing, please include in as well, accompanied by the DTA plots.

For the PZC plots, please replot initial pH and difference between initial and final pH, this would be visually clearer for readers.

.

Conclusion

Try to make the conclusion is more concise

Author Response

Author Response to Reviewer Comments on MS ID:  surfaces-1921009

Ternary metal-alginate-chitosan composites for controlled uptake of methyl orange

The author comments are listed below in a point-by-point fashion, as listed in blue font below.

Reviewer #3 Comments

Overall Comment

The paper entitled “Ternary metal-alginate-chitosan composites for controlled uptake of methyl oarnge”, showed some promising findings. Nonetheless, authors need to address some comments before ready for acceptance.

Abstract

Abstract is fine, however authors should include some numerical findings in for the physicochemical properties of the adsorbent in this case. Line 16 – 20 sounded oddly written. Why Fe-TCM-N showed such a low result, can you include some explanation there?

 Response:  The abstract was revised accordingly.

Introduction

Do add in the acute and long term health effects on human health in the writeup.

 Response:  The Introduction was revised accordingly.

Line 24-39, definition of dyes and adsorption are too vague in this case. Please refer to the manuscripts below for improvement

  1. Engineered macroalgal and microalgal adsorbents: Synthesis routes and adsorptive performance on hazardous water contaminants

https://www.sciencedirect.com/science/article/abs/pii/S0304389421018896

  1. Efficiency of various recent wastewater dye removal methods: A review

https://www.sciencedirect.com/science/article/abs/pii/S2213343718303695

Some in-text citations are missing the year info.

Response: We agree with the reviewer that the revisions added more context for the importance of MO removal and the benefits of adsorptive removal. The revised sections in the introduction were aimed at remaining concise and make brief reference to both suggested references as they are pertinent to this research.

Methodology

First of all, sub headings are missing for writeup

Response: The sub-headings have been added.

I have through the document for PZC measurement, why was the testing solution changed from calcium chloride to water? Based on what basis?

Response: The PZC was measured in NaCl solution (adapted from 0.005 M CaCl2 to 0.01 M NaCl). The rational was to avoid any potential adsorption of calcium onto alginate. We thank the reviewer for the detail oriented question and apologise for the oversight. We mention it now in the manuscript to avoid confusion.

For the preparation of composite materials, is this the first research work done to synthesize in such a manner|? If no, please include the ref.

Response: The synthetic procedure itself is based on previous work. The original synthetic path outlined by Kumar et al. and the subsequent work by Steiger et al. were cited, but the relevant section in “Materials and Methods” was added for clarification.

Generally, there are so many isotherms available, why particularly SIPS isotherm was chosen? Has the data been fitted with other isotherm as well, please include.

Response: Originally, Sips was chosen as it can account for both Langmuir of Freundlich behavior based on the versatility of the model. The adsorption follows the Langmuir isotherm model and the quality of the fits were improved according to the reviewer suggestions.

Results and Discussion

First of all, sub headings are missing for writeup

Response: Sub-headings have been added.

For the TGA, the thermograms are missing, please include in as well, accompanied by the DTA plots.

 Response: We have heeded the suggestion and revised the section accordingly.

For the PZC plots, please replot initial pH and difference between initial and final pH, this would be visually clearer for readers.

Response: We agree with the reviewer that this plot enhances the clarity and thus changed the plot accordingly.

.

Conclusion

Try to make the conclusion is more concise

Response: The conclusion has been revised to address the reviewer concern.

In summary, the Authors acknowledge Reviewer #3 for the constructive and insightful comments, along with the opportunity to improve the overall quality of the manuscript submission.

Reviewer 4 Report

Three different ternary metal composites (TMCs) were synthesized and their characteristics were studied; these TMCs were made from iron nitrate, aluminum nitrate, and copper nitrate, respectively (Fe-TMC-N, Al-TMC-N, Cu-TMC-N). Elements such as X-ray photoelectron spectroscopy (XPS), infrared (IR) spectroscopy, and an approximation of the point-of-zero-charge were used to characterize TMCs (PZC). Thermogravimetric analysis was used to study heat degradation (TGA). All things considered, the results of the materials characterization lend credence to the synthetic synthesis of the TMCs, where the surface charge is mostly determined by the metal nitrate precursor species. The surface chemistry of TMC materials was examined using methylene blue (MB) as a dye. Al-TMC-N and Cu-TMC-N showed little MB dye adsorption, whereas Fe-TMC-N showed high MB uptake (26 mg/g) at equilibrium. Pseudo-second order kinetics was observed for the equilibrium adsorption capacities of MO on Al-TMC-N (410 mg/g), Cu-TMC-N (384 mg/g), and Fe-TMC-N (41 mg/g), respectively. The adsorption of MOs is higher in Cu- or Al-based materials than in Fe-based TMCs, and iron inclusion in the structure makes them less appropriate for anionic dye removal.

What I regret is the absence of MEB and XPS characterization. In addition, a detailed isotherm and kinetics modeling are welcome: Intraparticule diffusion kinetic model has to be included in the study. Because Sipps' model doesn't give absolute satisfaction, you must try Langmuir and Freundlich models. I think that the Freundlich model will represent your data with high accuracy.

Author Response

Author Response to Reviewer Comments on MS ID:  surfaces-1921009

Ternary metal-alginate-chitosan composites for controlled uptake of methyl orange

The author comments are listed below in a point-by-point fashion, as listed in blue font below.

Reviewer #4 Comments

Three different ternary metal composites (TMCs) were synthesized and their characteristics were studied; these TMCs were made from iron nitrate, aluminum nitrate, and copper nitrate, respectively (Fe-TMC-N, Al-TMC-N, Cu-TMC-N). Elements such as X-ray photoelectron spectroscopy (XPS), infrared (IR) spectroscopy, and an approximation of the point-of-zero-charge were used to characterize TMCs (PZC). Thermogravimetric analysis was used to study heat degradation (TGA). All things considered, the results of the materials characterization lend credence to the synthetic synthesis of the TMCs, where the surface charge is mostly determined by the metal nitrate precursor species. The surface chemistry of TMC materials was examined using methylene blue (MB) as a dye. Al-TMC-N and Cu-TMC-N showed little MB dye adsorption, whereas Fe-TMC-N showed high MB uptake (26 mg/g) at equilibrium. Pseudo-second order kinetics was observed for the equilibrium adsorption capacities of MO on Al-TMC-N (410 mg/g), Cu-TMC-N (384 mg/g), and Fe-TMC-N (41 mg/g), respectively. The adsorption of MOs is higher in Cu- or Al-based materials than in Fe-based TMCs, and iron inclusion in the structure makes them less appropriate for anionic dye removal.

What I regret is the absence of MEB and XPS characterization. In addition, a detailed isotherm and kinetics modeling are welcome: Intraparticule diffusion kinetic model has to be included in the study. Because Sipps' model doesn't give absolute satisfaction, you must try Langmuir and Freundlich models. I think that the Freundlich model will represent your data with high accuracy.

Response: We appreciate the feedback and originally intended (for brevity) to not show detailed XPS characterization. This has been remedied where results for iron, copper and aluminium narrow scans are now included.

Furthermore, we have heeded the advice to employ the Langmuir and Freundlich isotherm models. The paragraphs mentioning equilibriums adsorption capacity have now been updated with the values obtained by the Langmuir model as it resulted in better fits compared with Freundlich.

SEM/MEB images have previously been shown for such copper materials by Udoetok et al.1 as well as Kumar et al.2, which does not specifically addresses the proposed knowledge gap on oxidation state, present metal species and its effect on MO adsorption. Hence, the authors suggest that the provided IR and XPS characterization suffice.

The IPD model has now also been included and we refer to the revised manuscript.

The authors have revised the manuscript based on the suggestion and furthermore added some context and toxicological effects of MO in the introduction. We thank the reviewer for the important feedback which aided in improving the manuscript.

In summary, the Authors acknowledge Reviewer #4 for the constructive and insightful comments, along with the opportunity to improve the overall quality of the manuscript submission.

Round 2

Reviewer 3 Report

Response: The PZC was measured in NaCl solution (adapted from 0.005 M CaCl2 to 0.01 M NaCl). The rational was to avoid any potential adsorption of calcium onto alginate. We thank the reviewer for the detail oriented question and apologise for the oversight. We mention it now in the manuscript to avoid confusion.

Comment: It should be written as fixed volume of 25 mL of 0.01 M NaCl, to….

Response: Originally, Sips was chosen as it can account for both Langmuir of Freundlich behavior based on the versatility of the model. The adsorption follows the Langmuir isotherm model and the quality of the fits were improved according to the reviewer suggestions.

Comment: The inclusion of isotherms were satisfactory. However, try to include the ref, for the original isotherms being reference, as a form of appreciation and respect, for Sip, Langmuir and Freundlich isotherms.

Furthermore, kindly also include how the model fitting was done? Using what software?

MO Dye adsorption.

Comment:

Table 1 should also be provided with the isotherm parameters of Freundlich and Sip, as well as their R2, for easier comparison for readers as well.

For Cu-TMC-N, the low R2 is unacceptable in this case, < 0.80.

Conclusion.

In the conclusion, it is best not to include any references. Try to be direct and concise.

Author Response

Reviewer reports on Manuscript ID: surfaces-1921009 – round 2

Reviewer #3 – round 2

Response: The PZC was measured in NaCl solution (adapted from 0.005 M CaCl2 to 0.01 M NaCl). The rational was to avoid any potential adsorption of calcium onto alginate. We thank the reviewer for the detail oriented question and apologise for the oversight. We mention it now in the manuscript to avoid confusion.

Comment: It should be written as fixed volume of 25 mL of 0.01 M NaCl, to….

Response: We have changed the wording accordingly in the revised Manuscript.

Response: Originally, Sips was chosen as it can account for both Langmuir of Freundlich behavior based on the versatility of the model. The adsorption follows the Langmuir isotherm model and the quality of the fits were improved according to the reviewer suggestions.

Comment: The inclusion of isotherms were satisfactory. However, try to include the ref, for the original isotherms being reference, as a form of appreciation and respect, for Sip, Langmuir and Freundlich isotherms.

Response: We included the original references for Langmuir, Sips and Freundlich models, as recommended.

Furthermore, kindly also include how the model fitting was done? Using what software?

Response: Origin 2021 Academic Version was used, in addition to MS excel (Office 365 Apps Enterprise version).

MO Dye adsorption.

Comment:

Table 1 should also be provided with the isotherm parameters of Freundlich and Sip, as well as their R2, for easier comparison for readers as well. For Cu-TMC-N, the low R2 is unacceptable in this case, < 0.80.

Response: We now included the fitting parameters in the Supporting Materials (cf. Table S2 & S3) based on the reviewers constructive comment. A revised Langmuir model fitting was carried out by removing outlier data in the linear region (steep slope) for the Cu-TMC-N material. This results in an overall improved fit, as evidenced by reduced the error and a higher value of R² = 0.87, as outlined in Table 1 of the revised Manuscript.

Conclusion.

In the conclusion, it is best not to include any references. Try to be direct and concise.

Response: The conclusion section was edited for clarity and conciseness. The cited references provide key independent support the conclusions drawn in this study so we feel it is important to retain the citations.

The authors values the constructive and helpful criticism provided by Reviewer #3, which has led to further improvements to the manuscript.

Reviewer 4 Report

Accept as it is

Author Response

Reviewer #4 – round 2

Accept as it is

Response:  The authors appreciate the constructive comments of Reviewer #4 and for the time for reviewing this manuscript.
